# Sleep Polygenic Risk Score Is Associated with Cognitive Changes over Time

**DOI:** 10.3390/genes13010063

**Published:** 2021-12-27

**Authors:** Angeliki Tsapanou, Niki Mourtzi, Sokratis Charisis, Alex Hatzimanolis, Eva Ntanasi, Mary H. Kosmidis, Mary Yannakoulia, Georgios Hadjigeorgiou, Efthimios Dardiotis, Paraskevi Sakka, Yaakov Stern, Nikolaos Scarmeas

**Affiliations:** 1Columbia University Irving Medical Center, Cognitive Neuroscience Division, New York, NY 10032, USA; ys11@columbia.edu; 21st Neurology Clinic, Department of Social Medicine, Psychiatry and Neurology, Eginition Hospital, National and Kapodistrian University of Athens, 11528 Athens, Greece; nikimourtzi23@gmail.com (N.M.); scharissis@gmail.com (S.C.); alhatzi@gmail.com (A.H.); e.ntanasi@hotmail.com (E.N.); 3Laboratory of Cognitive Neuroscience, School of Psychology, Aristotle University of Thessaloniki, 54124 Thessaloniki, Greece; kosmidis@psy.auth.gr (M.H.K.); ns257@cumc.columbia.edu (N.S.); 4Department of Nutrition and Dietetics, Harokopio University, 17671 Athens, Greece; myianna@hua.gr; 5School of Medicine, University of Thessaly, 41334 Larissa, Greece; gmhadji@med.uth.gr (G.H.); edar@med.uth.gr (E.D.); 6Athens Alzheimer’s Association, 11636 Athens, Greece; vsakka@ath.forthnet.gr

**Keywords:** sleep, polygenic risk score, cognition

## Abstract

Sleep problems have been associated with cognition, both cross-sectionally and longitudinally. Specific genes have been also associated with both sleep regulation and cognition. In a large group of older non-demented adults, we aimed to (a) validate the association between Sleep Polygenic Risk Score (Sleep PRS) and self-reported sleep duration, and (b) examine the association between Sleep PRS and cognitive changes in a three-year follow-up. Participants were drawn from the Hellenic Longitudinal Investigation of Aging and Diet (HELIAD). A structured, in-person interview, consisting of a medical history report and physical examination, was conducted for each participant during each of the visits (baseline and first follow-up). In total, 1376 participants were included, having all demographic, genetic, and cognitive data, out of which, 688 had at least one follow-up visit. In addition, an extensive neuropsychological assessment examining five cognitive domains (memory, visuo-spatial ability, attention/speed of processing, executive function, and language) was administered. A PRS for sleep duration was created based on previously published, genome-wide association study meta-analysis results. In order to assess the relationship between the Sleep PRS and the rate of cognitive change, we used generalized estimating equations analyses. Age, sex, education, *ApolipoproteinE-ε4* genotype status, and specific principal components were used as covariates. On a further analysis, sleep medication was used as a further covariate. Results validated the association between Sleep PRS and self-reported sleep duration (B = 1.173, E-6, *p* = 0.001). Further, in the longitudinal analyses, significant associations were indicated between increased Sleep PRS and decreased visuo-spatial ability trajectories, in both the unadjusted (B = −1305.220, *p* = 0.018) and the adjusted for the covariates model (B = −1273.59, *p* = 0.031). Similarly, after adding sleep medication as a covariate (B = −1372.46, *p* = 0.019), none of the associations between Sleep PRS and the remaining cognitive domains were significant. PRS indicating longer sleep duration was associated with differential rates of cognitive decline over time in a group of non-demented older adults. Common genetic variants may influence the association between sleep duration and healthy aging/cognitive health.

## 1. Introduction

Changes in sleep pattern follow aging, with specific sleep problems such as sleep fragmentation and daytime sleepiness occurring frequently in the elderly [1,2]. The circadian system and sleep homeostatic mechanisms become less robust with normal aging [1]. Sleep problems have been negatively associated with cognition, both cross-sectionally and longitudinally. In a previous study from our group, both poor sleep quality and long sleep duration were associated with poor memory performance in a group of older adults [3,4]. In a different study, we also showed that daytime sleepiness was associated with diminished cognitive performance in older adults. In a three-year follow-up, increased daytime sleepiness and sleep inadequacy were further associated with incident dementia [5,6].

Genetic variation in specific genes has been related to sleep regulation (i.e., *CLOCK*, *PER*, *BMAL1*), mostly associated with insomnia, circadian rhythms, and sleep homeostasis [7]. Critically, single nucleotide polymorphisms (SNP) in sleep-regulated genes have also been associated with cognitive phenotypes such as memory formation, consolidation, and cognitive alertness [7,8]. Similarly, the rhythmic expression of specific sleep genes (*BMAL1*, *CRY1*, and *PER1*) is also found to be disturbed in neurodegenerations such as mild cognitive impairment (MCI) and Alzheimer’s disease (AD) dementia [9]. Among the genes associated with sleep regulation is *Apolipoprotein E* (*APOE*-ε4), a major risk factor for late-onset AD [10]. Interestingly, our research in a large group of non-demented older adults indicated that *APOE*-ε4 carriers reported less snoring and subjective sleep apnea, even after controlling for multiple covariates [11,12]. A complex etiology combining environmental and genetic causes could affect the association between sleep and such genes, leaving space for more research in the field. Specific molecular clocks in different brain regions, their circadian phases, and their anatomical relationships may contribute to our understanding about the mechanisms of interaction between sleep and cognition [7]. Dashti and colleagues [13] recently reported 78 loci significantly associated with self-reported habitual sleep duration through genome-wide association analysis (GWAS) in 446,118 adults of European ancestry. In a different sample of a specific age-group of older adults [14], a polygenic risk score (PRS), derived from genetic variants associated with chronotype and sleep duration, identified a link between shorter sleep latency and higher levels of visuospatial ability, processing speed, and verbal memory. Based on the Dashti et al. study, we found that sleep duration’s polygenic score was associated with cognitive performance in a group of cognitively healthy adults aged 20–80 years old [15].

Although there is some research on specific sleep genes and cognition, there is no current longitudinal study, to our knowledge, regarding sleep PRS and cognitive changes over time in non-demented older adults. The aims of the current work were, in a large group of non-demented older adults, to: a) validate the association between sleep duration PRS (Sleep PRS) and self-reported sleep duration measure, and b) examine the association between Sleep PRS and cognitive changes in a three-year follow-up.

## 2. Methods

Participants were drawn from the Hellenic Longitudinal Investigation of Aging and Diet (HELIAD). HELIAD is a population-based, multidisciplinary, collaborative study designed to estimate the prevalence and incidence of MCI, AD, and other types of dementia, as well as other neuropsychiatric conditions of aging in the Greek population. All participants are aged 65 years or older. All participants’ ethnicity was white only. The study includes several demographic, medical, social, environmental, clinical, nutritional, and neuropsychological determinants, as well as the lifestyle activities of each participant. Follow-up occurs at approximately three-year intervals, with the same evaluation as the baseline taking place. HELIAD has been approved by the ethics institutional review boards of the National and Kapodistrian University of Athens (Eginition hospital) and University of Thessaly, and all volunteers gave written informed consent prior to their participation. More detailed information about the study can be found in previously published work [16].

Clinical/neuropsychological evaluation: A structured, in-person interview, consisting of a medical history report and physical examination, was conducted for each participant during each of the visits. All participants provided information about their current health status, any neurological conditions, past medical problems, surgeries they might have been through, hospitalizations, and any use of medication. They also answered valid questionnaires about their daily activities, physical exercise, and diet. We used the Blessed Dementia Scale in order to examine any change in their self-care habits, activities, and physical as well as cognitive function [17]. In addition, the Lawton Instrumental Activities of Daily Living was used in order to examine the capacity of each participant to independently perform daily activities [18].

In addition, an extensive neuropsychological battery examining five cognitive domains (memory, visuo-spatial ability, attention/speed of processing, executive function, and language) was administered. More precisely, cognitive function was evaluated by neuropsychologists through a comprehensive neuropsychological assessment of all major cognitive domains: orientation (Mini-Mental State Exam) [19], non-verbal and verbal memory (Medical College of Georgia Complex Figure Test -MCG) [20], Greek Verbal Learning Test [21], language (semantic and phonological verbal fluency) [22] was assessed with subtests of the Greek version of the Boston Diagnostic Aphasia Examination short form, namely, the Boston Naming Test (short form), and selected items from the Complex Ideational Material Subtest to assess verbal comprehension and repetition of words and phrases [23], visuo-perceptual ability (Judgment of Line Orientation [24,25] abbreviated form; MCG Complex Figure Test copy condition, Clock Drawing Test [26]), attention and information processing speed (Trail Making Test–TMT) [27], executive functioning (TMT–Part B; verbal fluency; Anomalous Sentence Repetition; Graphical Sequence Test; Motor Programming [20]; months forwards and backwards), and a gross estimate of intellectual level (a Greek multiple choice vocabulary test) [28].

First, z-scores were derived for each test, using the means and standard deviations (SD) calculated from the scores of the nondemented participants of the HELIAD study. The z-scores were calculated by subtracting the mean score from the individual score, and dividing by the standard deviation. Subsequently, individual neuropsychological test scores were grouped based on a priori neuropsychological knowledge of the particular cognitive functions that each test primarily examines, to produce an average domain score for memory, language, attention speed, and executive and visual-spatial functioning. A higher z-score indicates better cognitive performance.

Sleep questionnaire: Sleep quality was assessed using the sleep scale from the Medical Outcomes Study (MOS-SS) [29]. In order to examine sleep duration, we used the following question: “On the average, how many hours did you sleep each night during the past four weeks? Write in number of hours and minutes per night.” The final variable used was the sum of the total duration calculated in minutes.

Sleep medication: In order to control for sleep medication use, in further analysis, we created a variable including the following substances: hypnotics, narcotics, antipsychotics, anticholinergics, and phenobarbital. The variable was used dichotomously, with 1 as the use of at least one substance.

Diagnosis: The diagnosis of the clinical/cognitive status of each participant was reached through diagnostic consensus meetings of all the researchers and main investigators, both neurologists and neuropsychologists.

Genotyping: Genome-wide genotyping was performed for 1446 individuals at the facilities of the “Centre National de Recherche en Génomique Humaine” (Evry, Essonne, France) using the Illumina Infinium Global Screening Array (GSA, GSAsharedCUSTOM_24 + v1.0), as part of the European Alzheimer DNA biobank (EADB) project. A detailed description of the EADB genotyping, quality control (QC), and imputation can be found elsewhere [30]. In summary, variants included in the marker list for removal, provided by Illumina, or variants not uniquely aligned in the GRCh37 genome were excluded for further analysis. Moreover, variant intensity QC was conducted for all autosomal variants, according to established thresholds [31].

Next, we performed sample quality control using PLINK v1.9 software [32,33,34]. Specifically, samples with missingness >0.05, sex inconsistencies, or with a heterozygosity rate that deviated more than ±6 SD from the mean, were excluded. To identify population outliers, we ran principal component analysis (PCA), using as a reference dataset the population of 1000 Genome (phase 3), and we projected the combined dataset (1000 GP3 samples and the EADB samples) onto two dimensions using the flashPCA2 software [35]. To control for cryptic relatedness, we excluded one individual from each pair of samples with a kinship coefficient more than 0.125 (cut-off for third-degree relatives), yielding a final sample size of 1251 unrelated individuals. Regarding quality controls of variants, we excluded variants showing a missingness >0.05 in at least one genotyping center or having a differential missingness test *p* < 10^−10^. The Hardy–Weinberg equilibrium test (*p* < 10^−6^) was performed only in controls.

Imputation: To improve the accuracy of imputation, we compared the frequencies of variants (chi-square test) against two reference panels, the population of the Haplotype Reference Consortium r1.1 (HRC) [36], excluding samples from 1000 genomes as well as the Finnish and the non-Finnish population of Genome Aggregation Database v3 (gnomAD) [37]). Variants showing a x^2^ > 3000 in both HRC and gnomAD or a x^2^ > 3000 in one reference panel and not present in the other were excluded. Finally, genome-wide association studies (GWAS) were performed between controls across genotyping centers to assess frequency differences between genotyping centers, using the software SNPTEST [38], under an additive model and adjusting for PCs. Variants having a likelihood ratio test of *p* < 10^−5^ were excluded. Furthermore, we removed ambiguous variants with minor allele frequency (MAF) >0.4, and we kept only one copy of any duplicated variants, prioritizing the one with the lowest missingness.

All samples and variants passing the above QC metrics were imputed on a Michigan Imputation Server (v1.2.4) [39], using the TOPMed Freeze 5 reference panel. Phasing and imputation were performed using EAGLE v2.4 [40] and Minimac4 v4-1.0.2 software, respectively.

*Apolipoprotein E-ε4*: *ApoE*-ε4 has been associated with both sleep and cognition [11,12,41,42]. HELIAD participants were *APOE* genotyped based on previous publications [11,12,43]. *ApoE* genotypes were transformed into a dichotomous trait based on the number of *ApoE*-ε4 alleles: 0 if the individual does not carry any copy of the ε4 allele (non-ε4 carriers) or 1 if the individual carries 1 or 2 copies of the ε4 allele (ε4 carriers).

Polygenic Risk Score (PRS): Imputed dosages for a total of 5,611,082 SNPs with MAF > 0.05, call rate > 95%, and imputation quality score > 0.4 were converted to best-guess genotypes for PRS computation. The PRSice software [44] was utilized to construct PRSs for each individual, applying the clumping and thresholding (C + T) method. Different sets of SNPs were filtered in the HELIAD sample (i.e., target sample) by applying increasing *p*-value thresholds (PT) to the discovery GWAS meta-analysis summary statistics (0.0001, 0.001, 0.01, 0.05, 0.1, 0.2, 0.3, 0.4, 0.5, 1.0) [45], and appropriate linkage disequilibrium (LD)-based SNP clumping (SNP with *r*^2^ > 0.1 in 250 kb-windows were removed) was performed to ensure that only independent markers are included. PRS at each P_T_ was computed as the weighted sum of the risk-increasing alleles that each individual carries at each SNP locus, multiplied by the effect size for the reference allele on the basis of sleep duration GWAS meta-analysis results.

## 3. Statistical Analysis

Analyses were performed using SPSS 26 (SPSS, Chicago, IL, USA). Baseline characteristics of subjects were compared using the t-test or ANOVA models for continuous variables (i.e., age, education), and with the χ^2^ test for categorical baseline characteristics (i.e., sex). For the purposes of the current analysis, we excluded participants with the diagnosis of dementia at baseline.

Initially, we performed linear regression models to examine the association between the Sleep PRS and the self-reported sleep duration measure. In order to assess the relationship between Sleep PRS and the rate of cognitive decline, we used generalized estimating equations (GEE) models with a Gaussian probability distribution and an identity link function. The repeated measures for each subject were treated as a cluster [46]; potential within cluster correlations were considered by specifying an appropriate working correlation structure (exchangeable) based on the Quasi-likelihood under Independence Model Criterion (QIC). Each GEE model included the cognitive z-scores as the dependent variable, with time of follow-up (years), Sleep PRS, and an interaction term between time and Sleep PRS as predictors. A significant interaction term indicates differential rates of cognitive change over time as a function of the Sleep PRS. Models were initially unadjusted and then adjusted for: age at baseline visit, sex, years of education, the first two principal components (PCs) to control for potential population sub-structure, and *ApoE*-ε4. The PRS threshold that was used in the analyses was for P_T_ = 0.05. However, in order to increase the validity of our results, in further analysis we performed GEE analysis, examining the association between Sleep RPS and visuo-spatial ability in all PRS thresholds (0.0001, 0.001, 0.01, 0.05, 0.1, 0.2, 0.3, 0.4, 0.5, 1.0). In further analysis, sleep medication was used as a covariate as well.

## 4. Results

A total of 1376 non-demented participants were included in the study, with 688 having a mean of three-years follow-up (see Figure 1). They aged 73 mean years old (SD:5), with 8.4 mean years of education (SD:4.9), and were mostly women (see Table 1).

Results indicated a significant association between the Sleep PRS and the self-reported sleep duration measure (B = 1.173 × 10^−6^, *p* = 0.001), so that higher PRS indicated a longer sleep duration. Further, in the longitudinal analyses, there was a significant association between Sleep PRS and visuo-spatial ability in both the unadjusted (B = −1305.22, *p* = 0.018) and the adjusted for the abovementioned covariates’ model (B = −1273.59, *p* = 0.031) (see Table 2, Figure 2). In the further analysis, with sleep medication used as an additional covariate, results were similar for the association between Sleep PRS and visuo-spatial ability: B = −1372.46, *p* = 0.019). Higher Sleep PRS was associated with a more rapid decline of visuo-spatial ability overtime. None of the associations between Sleep PRS and the remaining cognitive domains were significant (see Table 2). We further created quartiles for the Sleep PRS and performed the same analysis, adjusted for the above-mentioned covariates. Participants in the highest quartile had greater decline in the visuo-spatial z-score per year of follow-up (Β = −0.066, *p* = 0.044).

Regarding the GEE analysis examining the association between Sleep RPS and visuo-spatial ability in other PRS thresholds, results confirmed the association: PRS P_T_ = 0.1 (Β = −1579.976, *p* = 0.024) and PRS P_T_ = 0.2 (Β = −2678.403, *p* = 0.008), after adding the same covariates in the model.

## 5. Discussion

Sleep function plays a significant role in normal aging and brain health. In the current study, we found that Sleep PRS was associated with visuo-spatial ability, so that genetic predisposition for a longer sleep duration is associated with a steeper decline in visuo-spatial cognitive performance. By replicating the association between Sleep PRS and self-reported measures in an independent cohort, we enhanced the GWAS results’ credibility of the original study, namely, that specific variation in sleep-associated SNPs is indeed associated with sleep duration. To our knowledge, our study is the first that goes a step further by examining the association between Sleep PRS and cognition over a three-year period, highlighting the importance of genetic variation in cognitive changes over time in non-demented older adults.

Our results are in accordance with and expand on existing literature, where, in a group of 70 to 76-year-old adults, longer daytime sleep duration was associated with slower speed of processing and steeper six-year decline in visuo-spatial reasoning and speed of processing [14]. Further, sleep problems have been associated with dysfunctions of the right parietal lobe, where visuo-spatial abilities reside [47,48], a possible reason why we see the association with the specific cognitive domain. Spatial visualization has been defined as the ability to mentally manipulate complex spatial information when several steps are necessary for successful completion of a spatial task [49]. Constructive and visuo-spatial abilities are complex fluid functions that decline with advancing age [50]. Specific tests measuring visuo-spatial ability can also differentiate normal from pathological aging [50]. Identifying associations between this cognitive ability and common genetic variations could shed light on how the brain functions.

Existing studies are connecting sleep function, usually measured by self-reported questionnaires or polysomnography/actigraphy, with cognition or dementia, in an attempt to address possible risk factors of cognitive decline. Our results go a step further and provide proof of the association between those SNPs previously associated with sleep duration and cognitive decline. Further, the fact that we did not include participants with dementia provides more evidence about the firm link between sleep and cognition. Based on our study, carriers of specific sleep-associated variants could possibly be more likely to develop cognitive decline over time, independent of demographic factors (age, sex, and education). As we have shown in previous publications, apart from sleep duration, daytime sleepiness (feeling drowsy or sleepy during the day) has been associated with both poor cognitive performance and increased incidence of dementia [5,6]. Thus, in future research, it would be helpful to examine which genes are shared between different sleep parameters of PRS, and further, any differences in their association with cognitive performance.

There is a high significance to our results, since genetic predisposition for sleep habits is revealed to play a role in the cognitive status of the elderly. The genetic influence of the rate of cognitive decline during normal aging could contribute to our understanding about the underlying mechanisms of neuronal function and about the well-established link between sleep homeostasis and brain function. The results of the current study could aid in the identification of people at higher risk for cognitive decline, early in life and before its manifestation. Further, our data could aid the discovery of genomic-driven drug treatments that take into account genetic variants associated with sleep, in order to maintain cognitive performance as we age, while, at the same time, potentially reducing the risk for age-related cognitive impairment.

On a previous study of our group, we found that higher sleep duration PRS was associated with good cognitive performance [15]. There are multiple possible reasons this occurs. Firstly, the two studies have included two different samples, with a similar but also different design, with also some of the neuropsychological tests being different, and different estimation of the cognitive domains. The use of a different reference study for the creation of the PRS might be another possible reason. Further, and very important, the sample of the first study included participants covering the whole adult age-range (20-80); with secondary analysis indicating that results were mostly driven by the young age group. Since the current sample includes older adults, this might be the main reason showing this difference. Another reason why this may occur is that at the first publication, the sample was very well educated, with a mean of 16 years of education, in contrary to the Greek sample which has a much lower educational level (M = 8.4), since education is highly associated with cognition. Lastly, the design of the two studies differs, with the first one being cross-sectional, while the second one being longitudinal.

To our knowledge, this is the first study examining Sleep PRS and cognitive performance over time in non-demented older adults aged 65 to 92 years old. Another strength is the extensive clinical and neuropsychological evaluation, resulting in well-diagnosed participants. By using genetics of sleep, we exclude possible noise of either subjective or even objective measurements. However, there are some limitations that should also be mentioned. The relatively small sample size is considered the main limitation of the study; further studies with larger sample sizes and including the whole adult age-range are needed to validate this novel finding. Given the role of sleep-associated genes and sleep duration in identifying cognitive functioning in older adults, for the creation of the Sleep duration PRS, objective sleep measures such as polysomnography or actigraphy would add more validity.

Sleep duration is a significant correlate of cognitive decline trajectories in visuospatial ability in older adults, before dementia. Identifying cross-phenotype associations of Sleep PRS may lay the groundwork for further investigations on the link between sleep genetics and possible neurodegenerative diseases. A genetic predisposition for cognitive decline based on sleep associated genes highlights the important role of early detection and treatment of sleep problems, in order to delay cognitive decline or even prevent dementia.

## Figures and Tables

**Figure 1 genes-13-00063-f001:**
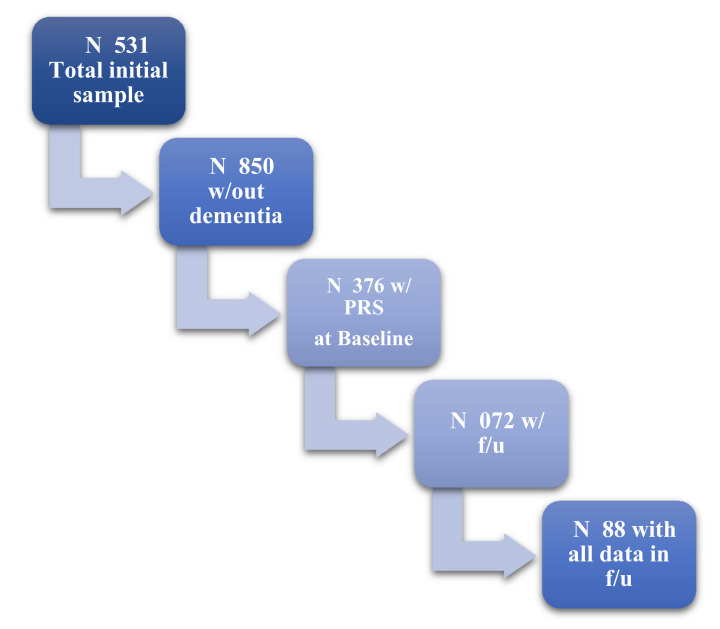
Flowchart of our sample based on available data and follow-up visits.

**Figure 2 genes-13-00063-f002:**
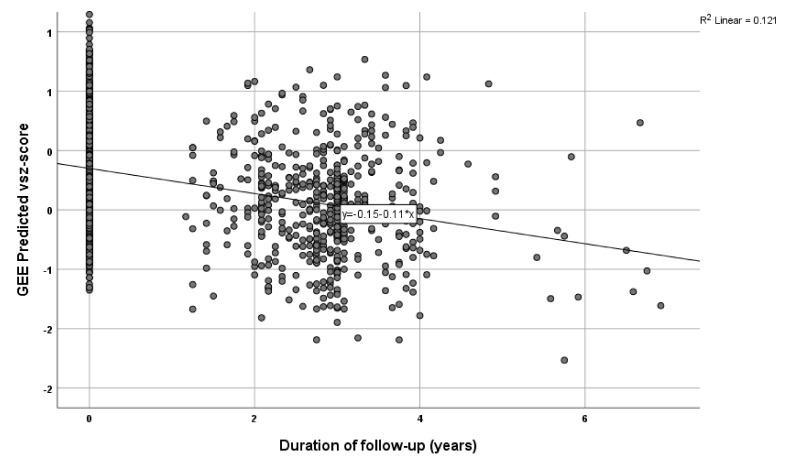
Graphic illustration of the cognitive decline in visuo-spatial ability over time (covariates are included).

**Table 1 genes-13-00063-t001:** Characteristics of the participants at baseline.

Characteristics	
Age, years Mean (SD)	73.1 (5)
Sex, women N (%)	1184 (60.5)
Education, years Mean (SD)	8.4 (4.9)
Cognitive domains	
Memory, Mean (SD)	−0.1342 (0.94031)
Executive, Mean (SD)	−0.1400 (0.78494)
Visuo-spatial, Mean (SD)	−0.1963 (0.92621)
Language, Mean (SD)	−0.0976 (0.90150)
Attention/Speed, Mean (SD)	−0.1840 (1.14814)
Duration of follow-up, years Mean (SD)	3.1 (0.9)
Total	1376

**Table 2 genes-13-00063-t002:** Association between Sleep PRS and cognitive changes. First, the unadjusted model, and then adjusted for age, sex, education, PC1, PC2, and *ApoE*-ε4.

Cognitive Ability	Β *	*p*
Memory	−551.69	0.173
	−759.8	0.083
Executive	−269.87	0.417
	−161.71	0.655
Visuo-spatial	−1305.22	0.018
	−1273.59	0.031
Language	30.947	0.936
	68.92	0.868
Attention/Speed	1253.22	0.058
	1184.24	0.086

* Values represent the beta coefficients of the interaction term: Sleep PRS*time.

## Data Availability

For data availability, please contact the corresponding author.

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
