# Peer review of "Sleep Polygenic Risk Score Is Associated with Cognitive Changes over Time"

_genes, 2021, doi:10.3390/genes13010063_

Round 1

Reviewer 1 Report

Recommendation for Genes:

Title: Sleep Polygenic Risk Score is associated with cognitive changes over time

Angeliki Tsapanou * , Niki Mourtzi , Socrates Charissis , Alexandros Hatzimanolis , Eva Ntanasi , Mary H Kosmidis , Mary Yannakoulia , Georgios Hadjigeorgiou , Efthimios Dardiotis , Paraskevi Sakka , Yaakov Stern , Nikolaos Scarmeas

This article using a polygenic risk score (PRS) associated with insomnia to predict cognitive decline in aged individuals without dementia. This is a well-powered study that identifies a significant association with sleep-PRS and longitudinal decreased visuo-spatial ability after adjusting for covariates. The longitudinal aspect of this dataset is a strength of the study. The authors were also able to validate their work as the sleep-PRS was also associated with longer sleep duration. Overall, this work provides an innovative investigation into the links between sleep and cognition in older adults. However, there are some concerns regarding this work that I would like to highlight below:

It is unclear why the authors chose this particular GWAS meta-analysis to use for the polygenic risk score [Ref 45; Line 186]. This is an insomnia GWAS, with 202 risk loci identified. Why did the authors not use the GWAS discussed in the introduction by Dashti et al [Ref 13,14], which the authors used in previous publications? Was there a reason they wanted to focus on an insomnia-PRS?

Can the authors comment on the disparate results presented here compared to their earlier study [Ref 15] which found that a higher sleep-PRS was associated with longer sleep duration, which is similar to what was found here, but also better cognition?

The inclusion of covariates is a strength of this study, but could the authors comment on if ethnicity or sleep medications were accounted for?

It is interesting that decreased visuo-spatial ability was the only cognitive characteristic that was predicted by the sleep-PRS. Can the authors discuss why the other cognitive characteristics would not be predicted by sleep-PRS?

Throughout the discussion, the authors present their results as “proof of the association between specific sleep genes” [Line 267] and “specific sleep-associated genes” [Line 270]. However, my interpretation of PRS would not be that specific. This PRS provides a measure of risk due to genetic variation, it cannot point to specific genes that you could call sleep-genes. Rather, they are genes with variants associated to insomnia. I think a clearer description of what these results tell us would be helpful in the discussion.

The authors comment that lifestyle changes could prevent cognitive decline [Lines 272-273] is also misleading as their data does not support this statement.

The authors need to define PT as thresholds in the paper.

Author Response

This article using a polygenic risk score (PRS) associated with insomnia to predict cognitive decline in aged individuals without dementia. This is a well-powered study that identifies a significant association with sleep-PRS and longitudinal decreased visuo-spatial ability after adjusting for covariates. The longitudinal aspect of this dataset is a strength of the study. The authors were also able to validate their work as the sleep-PRS was also associated with longer sleep duration. Overall, this work provides an innovative investigation into the links between sleep and cognition in older adults. However, there are some concerns regarding this work that I would like to highlight below:

It is unclear why the authors chose this particular GWAS meta-analysis to use for the polygenic risk score [Ref 45; Line 186]. This is an insomnia GWAS, with 202 risk loci identified. Why did the authors not use the GWAS discussed in the introduction by Dashti et al [Ref 13,14], which the authors used in previous publications? Was there a reason they wanted to focus on an insomnia-PRS?

Thank you for your comment. HELIAD is a different cohort than the one Dashti’s paper was used as a reference, thus, we created the Sleep PRS from scratch. We used the specific reference study (Jansen et al.) mostly because it included a much bigger sample size, thus, we considered it more accurate. Further, we chose the SNPs associated with sleep duration and not insomnia for the creation of the PRS. Lastly, other studies are also being prepared from our group based on the Jansen’s paper, thus, we wanted to be consistent.

Can the authors comment on the disparate results presented here compared to their earlier study [Ref 15] which found that a higher sleep-PRS was associated with longer sleep duration, which is similar to what was found here, but also better cognition?

Indeed, current results differ from the previously reported ones. However, there are multiple possible reasons this occurs. Firstly, the two studies have included two different samples, with a similar but also different design, with also some of the neuropsychological tests being different. Further, and very important, the sample of the first study included participants covering the whole adult age-range (20-80); with secondary analysis indicating that results were mostly driven by the young age group. Since the current sample includes older adults, this might be the main reason showing this difference. Another reason why this may occur is that at the first publication, the sample was very well educated, with a mean of 16 years of education, in contrary to the Greek sample which has a much lower educational level (M=8.4), since education is highly associated with cognition. Lastly, the design of the two studies differs, with the first one being cross-sectional, while the second one being longitudinal.  

In case the editor or the reviewer find it appropriate, we can add these comments in the manuscript.

The inclusion of covariates is a strength of this study, but could the authors comment on if ethnicity or sleep medications were accounted for?

Thank you for the very valuable comment. Ethnicity was not used as a covariate because in the specific group we do not have different ethnic groups, it is consisted of Whites only. This information is now added in the manuscript, in the Methods section, first paragraph: “All participants’ ethnicity was Whites only.”

Based on your comment, we now created a sleep medication variable and performed supplementary analysis regarding the association between Sleep PRS and cognition, with the results remaining unchanged.

We now made the corresponding changes throughout the manuscript:

Introduction: “On a supplementary analysis, sleep medication was used as a further covariate.”… “Similarly after adding sleep medication as a covariate (B=-1372.46, p=0.019).”

Methods section: “Sleep medication: In order to control for sleep medication use, in supplementary analysis, we created a variable including the following substances: hypnotics, narcotics, antipsychotics, anticholinergics, and phenobarbital. The variable was used dichotomously, with 1 as the use of at least one substance.”

Results: “In the supplementary analysis, with sleep medication used as an additional covariate, results were similar for the association between Sleep PRS and visuo-spatial ability: B=-1372.46, p=0.019).”

It is interesting that decreased visuo-spatial ability was the only cognitive characteristic that was predicted by the sleep-PRS. Can the authors discuss why the other cognitive characteristics would not be predicted by sleep-PRS?

Indeed, this is a very reasonable question that arises; we have now added in the Discussion section: “Our results are in accordance to and expand on existing literature, where, in a group of 70 to 76-year-old adults, longer daytime sleep duration was associated with slower speed of processing and steeper 6-year decline in visuo-spatial reasoning and speed of processing [14]. Further, sleep problems have been associated with dysfunctions of the right parietal lobe, where visuo-spatial abilities reside [47, 48], a possible reason the association occurs for the specific cognitive domain. Spatial visualization has been defined as the ability to mentally manipulate complex spatial information when several steps are necessary for successful completion of a spatial task [49]. Constructive and visuo-spatial abilities are complex fluid functions that decline with advancing age [50]. Specific tests measuring visuo-spatial ability can also differentiate normal from pathological aging [50]. Identifying associations between this cognitive ability and common genetic variations could shed light on how brain functions.”

Throughout the discussion, the authors present their results as “proof of the association between specific sleep genes” [Line 267] and “specific sleep-associated genes” [Line 270]. However, my interpretation of PRS would not be that specific. This PRS provides a measure of risk due to genetic variation, it cannot point to specific genes that you could call sleep-genes. Rather, they are genes with variants associated to insomnia. I think a clearer description of what these results tell us would be helpful in the discussion.

From the reference paper, we used the SNPs associated with sleep duration and not with insomnia. Thus, the PRS used was based on SNPs of genes highly associated with sleep duration. Based on your comment, we have now edited the discussion section of the manuscript accordingly.

The authors comment that lifestyle changes could prevent cognitive decline [Lines 272-273] is also misleading as their data does not support this statement.

We now retracted this comment.

The authors need to define PT as thresholds in the paper.

This information is now added in the manuscript, Methods section, Polygenic Risk Score (PRS): “Different sets of SNPs were filtered in the HELIAD sample (i.e., target sample) by applying increasing p-value thresholds (PT) to the discovery GWAS meta-analysis summary statistics (0.0001, 0.001, 0.01, 0.05, 0.1, 0.2, 0.3, 0.4, 0.5, 1.0) [45] and appropriate linkage disequilibrium (LD)-based SNP clumping (SNP with r2>0.1 in 250kb-windows were removed) was performed to ensure that only independent markers are included.”

Reviewer 2 Report

The main premise of this manuscript is to elucidate whether the sleep-associated genes (especially those linked with longer sleep duration) are implicated with a steeper decline in cognitive performance in a 3-year longitudinal study. Though this study was followed up for a limited time (3 years), it has significant potential in elucidating the role of Sleep PRS in cognitive decline in healthy geriatric individuals.

The replication of previous sleep GWAS results by examining the association between Sleep PRS and self-reported measures in an independent cohort is very encouraging. The study is well executed with an extensive clinical and neuropsychological evaluation with well-defined participants, which was accounted for in the data analysis. Such studies can provide some insight into the role of genes involved in inter-individual differences in sleep duration and cognition suggesting their role in multiple biological pathways.

Overall, the manuscript is well-written, data analysis is satisfactory based on which the authors have made substantial conclusions. I’ve few comments and remarks which are following-

Major comments:

  1. Is there any study on short sleepers for their association with cognitive functions? A couple of previous studies have reported an “inverted U-shaped” association between sleep duration and cognitive decline.
  2. Tsapanou et al previously reported that longer sleep duration is associated with better cognition (based on high PRS) in a healthy population aged 20-80 years, however, these results are very different when they looked at the non-demented aging population (65+ years). In the current study, the authors find the only association, which is significant, is between increased Sleep PRS and decreased visuospatial ability trajectories (all other associations with cognitive assessments were not significant). Is it due to different psychological measurements or due to the nature of this study i.e, longitudinal (current) vs cross-sectional (previous study)?
  3. Given the role of sleep-associated genes and sleep duration in identifying cognitive functioning in elderly individuals, rigorous large-sample studies, as well as more accurate sleep measurements (for eg, Polysomnography or actigraphy), are warranted as authors themselves point it out.
  4. Do other sleep parameters (in addition to Sleep duration) affect other cognitive functions or may not be associated with cognition at all? Some explanation in the discussion section would be helpful to the readers.
  5. Does this genetic disposition of longer sleep duration reveal its role in cognitive status in the younger population? How generalized is the role of altered sleep homeostasis in cognitive decline, if one examines different age groups? These points can be discussed in light of the results the authors have obtained.
  6. Sleep measurements were assessed every 4 weeks instead of relying on daily notes in the sleep diaries, the former method can be very subjective.
  7. Was only sleep duration polygenic score examined in the study that was found to be associated with cognitive performance? How about other sleep parameters?
  8. Was the quality of sleep assessed through the questionnaire? Several studies suggest sleep fragmentation is quite common in elderly individuals, which may result in lower cognitive performance in cognitively unimpaired elderly individuals. How about time to bed (Sleep Latency) affecting memory and stress on the following day?

Minor comments-

  1. Why were the PRS thresholds (p values) changed in the GEE analysis for examining the association between Sleep RPS and visuospatial ability?
  2. Line 134 and 135 should be joined.
  3. Line 151, correct the sentence to a past tense
  4. Line 162, Space between 1000genomes
  5. Line 169, p < 10-5 should be in superscript

Author Response

The main premise of this manuscript is to elucidate whether the sleep-associated genes (especially those linked with longer sleep duration) are implicated with a steeper decline in cognitive performance in a 3-year longitudinal study. Though this study was followed up for a limited time (3 years), it has significant potential in elucidating the role of Sleep PRS in cognitive decline in healthy geriatric individuals.

The replication of previous sleep GWAS results by examining the association between Sleep PRS and self-reported measures in an independent cohort is very encouraging. The study is well executed with an extensive clinical and neuropsychological evaluation with well-defined participants, which was accounted for in the data analysis. Such studies can provide some insight into the role of genes involved in inter-individual differences in sleep duration and cognition suggesting their role in multiple biological pathways.

Overall, the manuscript is well-written, data analysis is satisfactory based on which the authors have made substantial conclusions. I’ve few comments and remarks which are following:

Major comments:

  1. Is there any study on short sleepers for their association with cognitive functions? A couple of previous studies have reported an “inverted U-shaped” association between sleep duration and cognitive decline.

Most of the studies examining sleep duration and cognition report indeed an inverted U-shaped association. Scarce is the literature finding short sleep duration association with cognitive decline. A study from Dianne Keage et al., 2021 (What sleep characteristics predict cognitive decline in the elderly?) found that sleep ≤6h is a risk factor for incident dementia. To our knowledge, however, there is no such study regarding higher PRS being associated with shorter sleep duration.

  1. Tsapanou et al previously reported that longer sleep duration is associated with better cognition (based on high PRS) in a healthy population aged 20-80 years, however, these results are very different when they looked at the non-demented aging population (65+ years). In the current study, the authors find the only association, which is significant, is between increased Sleep PRS and decreased visuospatial ability trajectories (all other associations with cognitive assessments were not significant). Is it due to different psychological measurements or due to the nature of this study i.e., longitudinal (current) vs cross-sectional (previous study)?

As mentioned on the comments of the first reviewer, indeed, current results differ from the previously reported ones. However, there are multiple reasons this occurs. Firstly, the two studies have included two different samples, with a similar but also different design, with also some of the neuropsychological tests being different. Further, and very important, the sample of the first study included participants covering the whole adult age-range (20-80); with secondary analysis indicating that results were mostly driven by the young age group. Since the current sample includes older adults, this might be the main reason showing this difference. Another reason why this may occur is that at the first publication, the sample was very well educated, with a mean of 16 years of education, in contrary to the Greek sample which has a much lower educational level (M=8.4), since education is highly associated with cognition. Lastly, the design of the two studies differs, with the first one being cross-sectional, while the second one being longitudinal.  

In case the editor or the reviewer find it appropriate, we can add these comments in the manuscript.

  1. Given the role of sleep-associated genes and sleep duration in identifying cognitive functioning in elderly individuals, rigorous large-sample studies, as well as more accurate sleep measurements (for e.g., Polysomnography or actigraphy), are warranted as authors themselves point it out.

We now added this in the limitations section of the discussion: “However, there are some limitations that should be also mentioned. The relatively small sample size is considered the main limitation of the study; further studies with larger sample sizes and including the whole adult age-range are needed to validate this novel finding. Given the role of sleep-associated genes and sleep duration in identifying cognitive functioning in older adults, for the creation of the Sleep duration PRS, objective sleep measures such as polysomnography or actigraphy would add more validity.”

  1. Do other sleep parameters (in addition to Sleep duration) affect other cognitive functions or may not be associated with cognition at all? Some explanation in the discussion section would be helpful to the readers.

Based on your comment, we have now added in the Discussion section, third paragraph: “As we have shown on previous publications, apart from sleep duration, daytime sleepiness (feeling drowsy or sleepy during the day) has been associated with both poor cognitive performance and increased incidence of dementia (Tsapanou et al., 2015, Tsapanou et al., 2016). Thus, in future research, it would be helpful to examine which genes are shared between different sleep parameters PRS, and further, any difference with their association with cognitive performance.”  

  1. Does this genetic disposition of longer sleep duration reveal its role in cognitive status in the younger population? How generalized is the role of altered sleep homeostasis in cognitive decline, if one examines different age groups? These points can be discussed in light of the results the authors have obtained.

Unfortunately, the current study includes older adults only (aged 65 years or older). Thus, we do not have the ability to perform such analyses in younger groups, although it would be indeed interesting to examine how sleep homeostasis changes as we age. We also added this in the limitation section as mentioned above.

  1. Sleep measurements were assessed every 4 weeks instead of relying on daily notes in the sleep diaries, the former method can be very subjective.

Sleep measurements referred to the last 4 weeks of the time completed the questionnaire. Indeed, this is quite a subjective measurement, however, in the current study, we focused the main analysis on the Sleep PRS and cognition, and not these self-reported sleep measurements. However, as mentioned in the limitations section before, for the initial examination of the association between the PRS and sleep outcome, objective sleep measures would be definitely more accurate.

  1. Was only sleep duration polygenic score examined in the study that was found to be associated with cognitive performance? How about other sleep parameters?

In the current study only sleep duration PRS was examined for the association with cognition, meaning, the specific, previously associated with sleep genes. That being said, in order to examine different sleep parameters PRS, we would have to create a different PRS, based probably on different genes. Future studies could definitely focus on a different sleep PRS and its association with cognition, this is now added in the Discussion. 

  1. Was the quality of sleep assessed through the questionnaire? Several studies suggest sleep fragmentation is quite common in elderly individuals, which may result in lower cognitive performance in cognitively unimpaired elderly individuals. How about time to bed (Sleep Latency) affecting memory and stress on the following day?

Unfortunately, the MOS-SS sleep questionnaire does not include information about Sleep Latency, thus, such data are not available for further analysis in the current study. Regarding sleep quality in total, which is measured by the MOS-SS, similarly to the previous comment, we would be able to create a different Sleep quality PRS, using a different reference study, and then examine any association with cognition.

 Minor comments-

  1. Why were the PRS thresholds (p values) changed in the GEE analysis for examining the association between Sleep RPS and visuospatial ability?

The results based on different thresholds are mentioned for visuo-spatial ability only because it was the only statistically significant association. The association between these thresholds and the rest of the cognitive domains were not significant. The two different PRS thresholds PT =0.1 and PT =0.2 are mentioned in the results added in the basic analysis.

This is now clearly expressed in the manuscript, Statistical analysis section: “PRS threshold that was used in the analyses was for PT =0.05. However, in order to increase the validity of our results, in further analysis we performed GEE analysis examining the association between Sleep RPS and visuo-spatial ability in all PRS thresholds as well (0.0001, 0.001, 0.01, 0.05, 0.1, 0.2, 0.3, 0.4, 0.5, 1.0).”

  1. Line 134 and 135 should be joined.
  2. Line 151, correct the sentence to a past tense
  3. Line 162, Space between 1000genomes
  4. Line 169, p < 10-5 should be in superscript

 The above comments are now corrected in the manuscript.
